# Relationship Between Level of Trimethylamine Oxide and the Risk of Recurrent Cardiovascular Events in Patients with Acute Myocardial Infarction

**DOI:** 10.3390/nu17101664

**Published:** 2025-05-14

**Authors:** Wenjun Ji, Bin Zhang, Jiahui Liu, Kaiyin Li, Jia Jia, Fangfang Fan, Jie Jiang, Xingang Wang, Yan Zhang

**Affiliations:** 1Department of Cardiology, Peking University First Hospital, Beijing 100034, China; jiwenjun619@126.com (W.J.); raisend0101@163.com (B.Z.); jiahui-liu@pku.edu.cn (J.L.); 2361011122@pku.edu.cn (K.L.); jiajia9985@163.com (J.J.); fang9020@126.com (F.F.); jiangjie417@vip.163.com (J.J.); 2Institute of Cardiovascular Disease, Peking University First Hospital, Beijing 100034, China; 3State Key Laboratory of Vascular Homeostasis and Remodeling, Peking University, Beijing 100871, China; 4NHC Key Laboratory of Cardiovascular Molecular Biology and Regulatory Peptides, Peking University, Beijing 100871, China

**Keywords:** trimethylamine oxide, choline, betaine, L-carnitine, coronary heart disease, recurrent adverse cardiovascular events, cohort study

## Abstract

**Background:** This study investigated the value of trimethylamine oxide (TMAO) and its precursors in secondary prevention for patients with acute myocardial infarction (AMI). **Methods:** We retrospectively enrolled patients diagnosed with AMI. The associations of TMAO and its precursors with endpoint events were estimated by Cox proportional hazards models. **Results:** During a median follow-up of 6.4 years, 319 (32.0%) major adverse cardiovascular event (MACE) occurred in the 996 patients enrolled. After adjusting for traditional risk factors, the risk of MACE, cardiac death, and recurrent MI increased by 28% (HR 1.28, 95% CI 1.10–1.49), 44% (HR 1.44, 95% CI 1.12–1.84), and 27% (HR 1.27, 95% CI 1.04–1.55), respectively, per one increment in ln-transformed TMAO. After adjustment for the levels of its precursors, the relationship between TMAO and MACE was still significant. Choline was associated with MACEs, all-cause mortality, cardiac death, and risk of recurrent MI after adjusting for the levels of the remaining metabolites, in addition to traditional risk factors. The overall ability to predict all-cause mortality was better for the choline model than for the TMAO model (continuous NRI 0.185, *p* = 0.007; IDI 0.030, *p* = 0.020). Mediation effect analysis showed that the mediating effect of TMAO on choline and the risk of all-cause mortality was 11.39% (95% CI 0.0209–0.2200, *p* = 0.016), suggesting the existence of a choline activity pathway that is independent of the TMAO pathway. **Conclusions:** TMAO and choline were associated with an increased risk of MACE in patients with AMI, and choline had better predictive power.

## 1. Background

Coronary artery disease (CAD) is a major public health problem that imposes considerable disease and economic burdens [1]. Identification of the risk of cardiovascular and cerebrovascular events and early intervention is clinically important in patients with CAD, especially in those with myocardial infarction (MI). Aside from traditional risk factors, the relationship between nutrients and CAD has been receiving considerable attention.

Trimethylamine oxide (TMAO), one of the gut microbial metabolites most relevant to vascular disease, is an oxidative amine derived from choline, betaine, and carnitine, which are abundant in various types of seafood, dairy products, egg yolk, and meat [2]. Processed by the gut flora, these nutrients are hydrolyzed by trimethylamine lyase to trimethylamine, which then passes through the portal system to the liver, where it is further oxidized by hepatic flavin monooxygenase 3 to form TMAO [2,3]. The results of mechanistic studies indicate the existence of pathways relating to reverse cholesterol transport [4], formation of lipid-laden macrophages [5], vascular inflammation [6], and platelet hyperreactivity [7].

An untargeted metabolomics study published in 2011 was the first to link TMAO and its precursors with cardiovascular disease (CVD) [5]. After adjusting for traditional cardiac risk factors, TMAO, choline, and betaine were associated with an increased risk of CVD. More recent studies have shown that elevated TMAO and choline concentrations are associated with an increased risk of CVD and all-cause mortality [8,9,10,11]. The circulating TMAO level is an independent predictor of high atherosclerotic load in patients with CAD [12]. Baseline TMAO levels in patients with ST-elevation myocardial infarction (STEMI) were shown to predict the 1-year risk of adverse cardiovascular events [13]; however, other researchers have reached different conclusions [14]. Furthermore, some studies have suggested that renal function plays an important role in the association between TMAO and the risk of cardiovascular events, but that the effect is no longer significant after adjustment for renal function [15,16]. In summary, there is controversy regarding the value of TMAO for secondary prevention, especially in patients with MI, and renal function may confound the relationship between TMAO and CVD.

It has been shown that the TMAO level remains associated with the risk of stroke even after adjusting for levels of TMAO precursor substances [17] and that high choline, betaine [18], and L-carnitine [4] levels are associated with a higher risk of future major adverse cardiovascular event (MACE) only in subjects with concomitant elevation of TMAO, suggesting that plasma TMAO is a major driver of this association. However, there are some conflicting reports. Shea et al. [19] found that choline was associated with an increased risk of CVD after adjusting for TMAO and betaine, whereas TMAO was not associated with an increased risk of CVD after similar adjustments.

Therefore, the significance of TMAO and metabolites in secondary prevention of coronary heart disease has not yet been fully elucidated. In this study, we explored the secondary prevention value of TMAO and its precursors in patients with acute myocardial infarction (AMI).

## 2. Methods

### 2.1. Study Population

The study population consisted of adult patients who were admitted to the Department of Cardiology at Peking University First Hospital with a diagnosis of AMI between January 2010 and December 2018. The following inclusion criteria were applied: (1) age older than 18 years, (2) fulfillment of diagnostic criteria for AMI, (3) written informed consent provided, and (4) availability of a blood sample. According to the third universal definition of myocardial infarction, a diagnosis of AMI is based on the presence of myocardial injury markers above the 99th percentile of the upper limit of the normal reference value on the patient’s first blood test at the time of admission to hospital or at 1 month before admission, with at least one of the following as evidence of ischemia: (1) symptoms typically associated with angina pectoris, such as chest pain, chest tightness, and radiating pain; (2) ischemic changes on an electrocardiogram, including ST-segment depression or elevation, T-wave inversion, new-onset left bundle branch block, and new-onset idiopathic Q-waves; (3) echocardiogram or cardiac nuclear magnetic resonance scan showing new-onset loss of surviving myocardium or abnormalities of ventricular wall motion; and (4) presence of intracoronary thrombus confirmed by coronary angiography. Patients with no biological samples available at baseline for the TMAO and related metabolite assay were excluded. The study was approved by the ethics committee of Peking University First Hospital and performed in accordance with the provisions of the Declaration of Helsinki. All participants provided written informed consent.

### 2.2. Collection of Baseline Data

The baseline data were collected from the electronic medical records system and entered into the study database using EpiData 3.1 software (Odense, Denmark). The baseline data included sex, age, type of MI, history of hypertension, diabetes, old MI, and stroke, smoking and alcohol consumption, and medication history, including any antihypertensive, hypoglycemic, or lipid-lowering agents taken during the 3 months before hospitalization. Standard secondary prevention therapy for coronary heart disease was defined as concomitant treatment with dual antiplatelet therapy, a statin, an angiotensin receptor inhibitor or angiotensin-converting enzyme inhibitor, or a beta-blocker at the time of hospital discharge.

### 2.3. TMAO Metabolites

Plasma TMAO, choline, betaine, and L-carnitine levels were determined by stable-isotope dilution liquid chromatography–tandem mass spectrometry. Blood samples were collected from the arterial sheath before coronary angiography in the cardiac catheterization room and centrifuged at 2700 rpm for 10 min at room temperature within 30 min of collection, after which the plasma was extracted and stored in a freezer at −80 °C. TMAO metabolite levels were determined as follows: protein precipitation as sample pretreatment using an Acquity UPLC^®^ BEH HILIC column (Waters, Milford, MA, USA) measuring 2.1 × 100 mm × 1.7 µm with 95% acetonitrile–water (containing 10 mmol/L ammonium acetate, 0.5% formic acid) as the mobile phase, a flow rate of 0.8 mL/min, an injection volume of 10 μL, and gradient elution. The ion source was electrospray ionization. The ionization mode was positive through multiple reaction monitoring. The quantification was performed by the internal standard calibration method using the Triple Quad 4500 system (Sciex, Framingham, MA, USA) as the detection instrument.

### 2.4. Laboratory Examination

Biological markers, including total cholesterol, triglycerides, low-density lipoprotein cholesterol (LDL-C), high-density lipoprotein cholesterol (HDL-C), and blood creatinine, were measured using the first values detected during hospitalization, whereas cardiac troponin I was measured using the peak value during hospitalization. These values were extracted from electronic medical records. The left ventricular ejection fraction (LVEF) measured on the last echocardiogram before discharge was used as an index of cardiac function. Echocardiography was performed, and images were acquired using a Vivid E9 diagnostic system (GE Healthcare, Chicago, IL, USA).

### 2.5. Cardiovascular Outcomes

The primary endpoint events in this study were MACE, a composite endpoint that included cardiovascular death, first non-fatal MI, and stroke. The secondary endpoint events were all-cause mortality, cardiovascular death, recurrence of MI, and first ischemic or hemorrhagic stroke (fatal or non-fatal). The endpoint data sources were obtained by matching the identification information for the study participants with external databases. Information on non-fatal MI and stroke events was obtained from the Beijing Municipal Health Commission–Beijing inpatient medical record system, which included diagnoses for the study participants on the home pages of all healthcare institutions in Beijing (non-military healthcare facilities) from the time of enrollment through to 31 December 2021. The diagnostic information on the home page for the hospitalized cases was based on the 6-digit (extended) code of the International Classification of Diseases, 10th Revision (ICD-10), as shown in Appendix A. The death information was obtained from the Chinese Center for Disease Control and Prevention–National Mortality Surveillance System, with a cut-off date of 31 December 2021. The disease code was based on the ICD-10 4-digit code (subheading) form, the details of which are shown in Appendix A.

To avoid missing endpoint events, information through 31 December 2021 was obtained by follow-up telephone call and entered into a pre-designed telephone follow-up form, which included whether death had occurred; the date and cause of death; whether hospitalization for heart disease, cerebrovascular disease, or other vascular disease occurred again after discharge from hospital; whether there was any visit to the emergency room for MI, cerebral infarction, or cerebral hemorrhage; and whether medications for secondary prevention of CAD were taken regularly. The endpoints were determined by two senior physicians who performed “back-to-back” double event determination. In the event of disagreement, a third physician confirmed the event independently. The three adjudicating physicians were unaware of the clinical data and the results of metabolite tests, such as TMAO tests.

### 2.6. Statistical Analysis

Continuous variables are expressed as the mean ± standard deviation or the median (interquartile range) as appropriate. Categorical variables are shown as the frequency (percentage). Continuous variables were compared between groups using analysis of variance or the Kruskal–Wallis H test. Categorical variables were compared between groups using the chi-squared test or Fisher’s exact test (if the theoretical number was <10).

Kaplan–Meier curves and log-rank tests were used to examine any differences in event-free survival between the three subgroups of TMAO and its precursors. The relationship between levels of plasma TMAO and its precursors and the risk of endpoint events was evaluated using univariable and multivariable Cox proportional hazard models. We performed Ln transformations on TMAO and its precursors as continuous variables. Model 1 was adjusted for sex; age; type of MI; history of hypertension, diabetes, MI, and stroke; history of percutaneous coronary intervention (PCI) and coronary artery bypass grafting (CABG); smoking and alcohol consumption; creatinine, triglyceride, HDL-C, LDL-C, and cardiac troponin I levels; LVEF; number of vessels; emergency PCI or elective PCI for this episode; discharge medication (aspirin, carbamazepine, statin, beta-receptor antagonist; and angiotensin-converting enzyme inhibitor/angiotensin receptor blocker). Model 2 was adjusted for the levels of the remaining three of the four metabolites, in addition to the factors in model 1.

Based on the results of the above tests, a model was constructed to compare positive metabolites with other variables in model 1. The predictive value of the model was assessed by calculating the C-index, continuous net reclassification improvement (NRI), and integrated discrimination improvement (IDI) values. The mediation effects were analyzed using the R mediation package.

Interaction terms were included in the multivariable Cox proportional risk regression models to analyze the modifying effects of the following factors on the associations of TMAO and its precursors with outcomes: sex, age (<65 years vs. ≥65 years), type of MI (STEMI vs. non-STEMI), hypertension, diabetes mellitus, old MI, history of PCI, history of CABG, stroke, history of smoking and alcohol consumption, LVEF (<50% vs. ≥50%), blood creatinine (<82.8 μmol/L vs. ≥82.8 μmol/L), LDL-C (<1.8 mmol/L vs. 1.8–2.5 mmol/L vs. ≥2.6 mmol/L), peak cardiac troponin I (<3.4 ng/mL vs. ≥3.4 ng/mL), number of vessels associated with the lesion, interventions, and standard coronary medications used for secondary prevention.

All data analyses were performed using Empower (R) (version: 2.0, http://www.empowerstats.com (accessed on 3 January 2025)) and R (version: 4.1.0, http://www.R-project.org (accessed on 4 January 2025)). All tests were two-tailed, and a *p*-value < 0.05 was considered statistically significant.

## 3. Results

### 3.1. Baseline Characteristics of the Study Participants

During a median follow-up of 6.4 years (interquartile range 4.3, 9.0), 319 (32.0%) MACEs occurred in the 996 patients enrolled in the study. The study participants were divided into those with MACEs (the MACE group) and those without MACEs (the non-MACE group) (Table 1). Compared with patients in the non-MACE group, those in the MACE group were older, had more comorbidities, such as hypertension, diabetes mellitus, stroke, old MI, and chronic kidney disease, and were more likely to have three-branch coronary lesions, but they had smaller proportions with emergency or elective coronary interventional therapy and standard secondary prevention therapy. TMAO, choline, betaine, and L-carnitine levels were significantly higher in the MACE group, as was the creatinine level, while the levels of total cholesterol, triglyceride, and LDL-C, as well as the LVEF, were lower than in the non-MACE group.

### 3.2. Associations of TMAO, Choline, Betaine, and L-Carnitine Levels with All Endpoint Events

Figure 1 shows the results of the smooth curve of log-transformed TMAO and precursor levels to MACE after adjustment for traditional cardiovascular risk factors. The risk of MACE increased with increasing choline and L-carnitine levels.

Figure 2 shows Kaplan-Meier curves of TMAO and precursor tertiles groups. The proportion of MACE significantly increased with the elevation of TMAO and choline levels, while betaine and L-carnitine did not show this trend.

Table 2 demonstrates the relationship between TMAO and each endpoint event. Per one increment in ln-transformed TMAO level, the risk of MACE, all-cause mortality, cardiac death, MI, and stroke were increased by 65% (hazard ratio [HR] 1.65, 95% confidence interval [CI] 1.49–1.82), 82% (HR 1.82, 95% CI 1.62–2.04), 90% (HR 1.90, 95% CI 1.63–2.22), 64% (HR 1.64, 95% CI 1.44–1.87), and 54% (HR 1.54, 95% CI 1.28–1.86), respectively. When further adjusted for age; sex; type of MI; hypertension; diabetes mellitus; old MI; previous PCI and CABG; smoking and alcohol consumption; creatinine, triglyceride, HDL-C, LDL-C, and peak cardiac troponin I levels; LVEF; number of diseased vessels; emergency PCI; elective PCI; and medications (model 1), the risks of MACE, cardiac death, and recurrent MI were increased by 28% (HR 1.28, 95% CI 1.10–1.49), 44% (HR 1.44, 95% CI 1.12–1.84), and 27% (HR 1.27, 95% CI 1.04–1.55), respectively, per one increment in ln-transformed TMAO. However, when the levels of the other three metabolites were further adjusted (model 2), only TMAO was significantly associated with MACE. The associations between the TMAO level and each endpoint event were explored further by categorizing the TMAO level into tertiles and using the bottom tertile as a reference; the *p*-trends for TMAO and MACE, recurrent MI, and stroke were significant in model 1. Compared with patients in the bottom tertile, those in the top tertile had an increased risk of MACE (HR 1.56, 95% CI 1.13–2.15). When the levels of the other three metabolites were further adjusted (model 2), the *p*-trend for TMAO and MACE was also significant.

As shown in Table 3, with the increase in ln-transformed choline, the risks of MACE, all-cause mortality, cardiac death, and recurrent MI were increased; the HR were 2.84 (95% CI 1.90–4.23), 3.39 (95% CI 2.12–5.40), 6.86 (95% CI 3.51–13.41), and 2.61 (95% CI 1.54–4.43), respectively (model 1). This relationship remained when the other three metabolites were adjusted further. The associations between choline and each endpoint event were similar to those reported above when further explored by categorizing the choline levels into tertiles and using the bottom tertile as a reference. In model 1, the adjusted HR for MACE, all-cause mortality, cardiac death, and recurrent MI for those in the top tertile were 1.91 (95% CI 1.38–2.65, *p*-trend < 0.001), 2.06 (95% CI 1.39–3.05, *p*-trend < 0.001), 5.19 (95% CI 2.64–10.19, *p*-trend < 0.001), and 1.90 (95% CI 1.23–2.94, *p*-trend = 0.003), respectively. The adjusted model 2 showed results consistent with those described above.

Appendix A demonstrates the relationship between betaine and each endpoint event. With the increase in ln-transformed betaine, the risks of all-cause mortality, cardiac death, and recurrent MI were increased, with respective HR of 1.57 (95% CI 1.00–2.48), 2.59 (95% CI 1.36–4.93), and 1.77 (95% CI 1.04–3.02) (model 1). The associations between betaine and each endpoint event were similar to those described above when further explored by categorizing the betaine levels into tertiles and using the bottom tertile as a reference. Compared with the bottom tertile, those in the top tertile had increased risks of all-cause mortality (HR 1.43, 95% CI 1.01–2.04, *p*-trend = 0.043) and cardiac death (HR 1.64, 95% CI 1.00–2.68, *p*-trend = 0.040). However, after further adjustment for the other three metabolites, the associations of the betaine level with all-cause mortality and cardiac death were no longer significant in the upper tertile. The relationship between L-carnitine and each endpoint event was not statistically significant in either model (Appendix A).

Figure 3 demonstrates the relationship between TMAO, choline, betaine, and L-carnitine and each endpoint event in brief.

### 3.3. Subgroup Analysis and Interaction Test

Next, we performed a subgroup analysis and interaction test to evaluate the relationship between TMAO, choline, betaine, L-carnitine, and MACE and potentially modifying factors, including age, sex, type of MI, hypertension, diabetes mellitus, previous MI, previous PCI and CABG, previous stroke, smoking and alcohol consumption, LVEF, blood creatinine, LDL-C, peak cardiac troponin I, number of diseased vessels, emergency and elective PCI, and standard medication for prevention. The relationship between the baseline plasma TMAO level and the risk of MACE was stronger in patients who were female (HR 1.59, 95% CI 1.23–2.06, *p*-interaction = 0.038), those who did not have hypertension (HR 1.65, 95% CI 1.30–2.10, *p*-interaction = 0.008), and those who did not consume alcohol (HR 1.44, 95% CI 1.21–1.72, *p*-interaction = 0.005) (Figure 4). The relationship between choline and MACE varied according to the number of diseased vessels, with a 272% (HR 3.72, 95% CI 1.47–9.38, *p*-interaction = 0.048) increase per one increment in ln-transformed choline level in patients with double-branch lesions (Appendix A). The blood creatinine level modified the association of the plasma betaine level with the risk of MACE, with a stronger association found in patients with a blood creatinine of <82.8 μmol/L than in those with a blood creatinine of ≥82.8 μmol/L (HR 2.64, 95% CI 1.34–5.20, *p*-interaction = 0.015).

### 3.4. Assessment of Forecasting Effectiveness and Analysis of Mediating Effects

Given that plasma TMAO and choline levels still had predictive value for endpoint events after adjusting for the remaining metabolites, we constructed separate prediction models for TMAO and choline with other variables of the prediction model. Figure 5 shows the time-dependent areas under the curve for TMAO, choline, and the various endpoint events. Furthermore, as shown in Table 4, the overall ability of the choline model to predict all-cause mortality was better than that of the TMAO model (continuous NRI = 0.185, *p* = 0.007; IDI = 0.030, *p* = 0.020). As shown in Figure 6, the mediating effect of TMAO on choline and the risk of all-cause mortality accounted for 11.39% (95% CI 0.0209–0.2200, *p* = 0.016). In contrast, for MACE, cardiac death, recurrent MI, and stroke events, average causal medication effects were not significant. However, average direct effects and total effects were significant, suggesting the existence of an activity pathway for choline that is independent of the TMAO pathway.

## 4. Discussion

This study had four main findings. First, after a median follow-up of 6.4 years, baseline TMAO and choline levels in patients with AMI were associated with the risk of MACE, cardiac death, and recurrent MI, both as continuous variables and categorical variables, after adjusting for possible confounding factors. Furthermore, choline was associated with all-cause mortality, while betaine was associated with the risk of cardiac death. Second, after further adjustment for the remaining three metabolites, TMAO remained correlated with MACE, while choline remained correlated with MACE, cardiac death, all-cause mortality, and risk of recurrent MI. Third, stratification and the interaction results showed that the baseline TMAO level was more strongly associated with MACE in women, patients without comorbid hypertension, and patients who did not consume alcohol. No factors were found to modify the association between choline and the risk of MACE, indicating that the relationship between choline and MACE is more stable. Fourth, the overall ability of the choline model to predict all-cause mortality was better than that of the TMAO model. Moreover, the mediation effect analysis showed that the mediation effect of TMAO on the risk of choline vs. the risk of all-cause mortality events accounted for 11.39% and that there may be activity pathways for choline that were not dependent on the TMAO pathway.

In a 3-year follow-up of 3903 patients who underwent non-emergency coronary angiography, the higher the plasma choline or betaine level, the higher the risk of MACEs (including future all-cause mortality, non-fatal MI, and non-fatal stroke), even after adjustment for traditional risk factors [18]. Notably, when TMAO was included in the model, the correlations of choline and betaine with the risk of MACE were significantly attenuated or disappeared. In light of this finding, an increasing number of studies have focused on assessment of TMAO and its precursors, particularly TMAO. However, previous studies have rarely included TMAO, choline, betaine, and L-carnitine together and have mostly assessed their roles separately. Our study used the strategy of adjusting for the remaining metabolites and performing mediating effects analyses to explore the independent effects of TMAO and its precursors on endpoint events. We found that after additional adjustment for the remaining three metabolites, the associations of choline with the risk of MACE, all-cause mortality, cardiac death, and recurrent MI and the association of TMAO with the risk of MACE remained robust. Similarly, a recent US cohort study in young adults (the CARDIA cohort) explored the associations of TMAO, choline, and betaine with the development of CVD over a 19-year period and found that choline was positively associated with the risk of CVD after adjusting for demographics, traditional risk factors for CVD, and the remaining two metabolites (i.e., TMAO and betaine); however, in the same adjusted analyses, TMAO and betaine were not associated with the risk of CVD [19]. Those findings suggest that choline may be a more important risk indicator than TMAO. However, a nationwide nested case–control study in China found that even after adjusting for TMAO precursors, higher levels of TMAO (rather than those of its precursors) were associated with an increased risk of stroke in a community-based population [17]. However, both of these studies were performed in community-dwelling populations, whereas our study was performed in patients with AMI. Another recent nationwide study that explored the associations of TMAO and its precursors with the prognosis in patients with AMI during a median follow-up of 739 days also found a relationship between the baseline TMAO level and MACE (including all-cause mortality, recurrent MI, and hospitalization for heart failure, stroke, or any revascularization) [20]. In contrast, our study included a longer period of follow-up and yielded more convincing results for MACE, defined as all-cause mortality, cardiac death, and recurrent MI,

In recent years, the impact of renal function as a potential mediator or contributor to the negative effects of TMAO has been controversial [21]. However, the cardiorenal toxicity of TMAO may be more pronounced in the presence of reduced renal function. A prospective cohort study of subjects with suspected CAD in Norway did not support the association of TMAO with all-cause, cardiovascular, or non-cardiovascular deaths after adjustment for estimated glomerular filtration rate [16]. Furthermore, the prospective CHS cohort study in the USA demonstrated that further adjustment for estimated glomerular filtration rate attenuated the association of TMAO with a higher rate of recurrence of atherosclerotic CVD [15]. However, in our study, after adjusting for the blood creatinine level, TMAO was still correlated with the risk of MACE, cardiac death, and recurrent MI, underscoring the important role of TMAO in the prognosis of infarction.

Our subgroup analysis showed a more significant relationship between the baseline TMAO level and the risk of MACE in female patients. Baranyi et al. similarly found that female patients tended to have a higher cardiovascular risk owing to a persistently higher TMAO level after infarction [22]. This finding may reflect a sex-related difference in TMAO levels, whereby female patients have a higher risk of adverse events post-infarction and need to be closely monitored. Furthermore, TMAO was more strongly associated with the risk of MACE in patients without comorbid hypertension and in those who did not consume alcohol. It is now widely accepted that moderate alcohol consumption may be beneficial to the heart, possibly because of a favorable effect on several surrogate endpoints of atherosclerosis (e.g., hemoglobin, coagulation and fibrinolytic factors, endothelin, and oxidative stress) [23]. This favorable effect may ameliorate the atherogenic effects of TMAO and may explain the effect of alcohol consumption on the relationship between TMAO and the MACE-modifying effects seen in our study. However, given that the amount and pattern of alcohol consumption were not assessed, our results can only be considered hypothesis-generating. Our study showed a more significant association between plasma TMAO and the risk of recurrent MACE in normotensive patients, which may reflect the fact that we included a population of patients with MI who had more complex comorbidities and concomitant medications. It has been reported that angiotensin-converting enzyme inhibitors increase urinary excretion of TMAO in mice [3]. In patients with CVD, the use of collateral diuretics is associated with increased plasma concentrations of TMAO [24]. Given the complexity of the various diseases and background drugs, more studies are needed to assess this relationship.

We found that the ability of choline to predict the risk of all-cause mortality was partially mediated by TMAO, as well as the existence of an activity pathway for choline that is not dependent on the TMAO pathway. As an essential nutrient in humans, in addition to its involvement in the metabolism of TMAO, cholinergic neurotransmission, and synthesis of phosphatidylcholine, its oxidized byproduct betaine can act as a methyl donor in the remethylation of homocysteine to methionine, which is a recognized risk factor for coronary heart disease because of its cytotoxic effects on the vascular endothelium [25]. Increased plasma homocysteine and subsequent vascular endothelial cell toxicity after methionine loading and low plasma choline and betaine levels diminish the ability of homocysteine to methylate to methionine, resulting in increased aberrant methylation of plasma homocysteine [26], which may increase atherosclerosis [27]. Although there is a large body of evidence suggesting that choline, as a TMAO precursor, is involved in the development of atherosclerotic CVD through the TMAO pathway, there are a number of conflicting reports [18,28,29]. These inconsistent findings may reflect the existence of additional activity pathways for choline, as well as differences in choline intake and plasma levels in different populations. Further mechanistic studies are needed to explore this relationship.

The findings of previous clinical studies regarding the prognostic value of TMAO and its precursors for each endpoint event have been inconsistent. In the present study, TMAO, choline, betaine, and L-carnitine were studied simultaneously in patients with AMI. In addition to adjusting for traditional confounders, the remaining metabolites were further adjusted, and mediation effect analyses were performed to assess the respective roles of TMAO and its precursors and to compare their efficacy. However, this study has some limitations. First, it had a single-center retrospective design, and the sample size was small. Large-scale multicenter studies in China are needed to validate our findings. Second, although we showed that choline has an activity pathway that is not dependent on TMAO, we did not include a mechanistic study, so further exploration of this observation is needed in the future. Third, dietary habits, gut microbiota composition, and adherence to post-discharge medications were not collected in the study, which might, to some extent, affect our interpretation of the results.

## 5. Conclusions

The correlations of plasma TMAO and choline levels with a poor prognosis in Chinese patients with AMI highlights the value of secondary prevention of CAD. The possibility that choline is a more important metabolite than TMAO requires further investigation.

## Figures and Tables

**Figure 1 nutrients-17-01664-f001:**
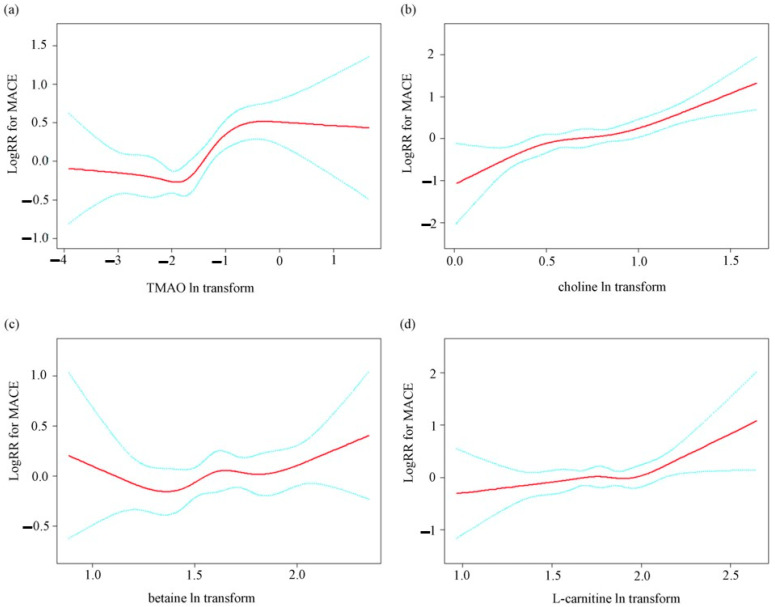
The smooth curve of log-transformed TMAO and precursor levels to MACE after adjustment for traditional cardiovascular risk factors. The red line represents the fitting line, and the blue line represents the 95% confidence interval: (**a**) TMAO, (**b**) choline, (**c**) betaine, (**d**) L-carnitine. Model 1 included sex; age; type of MI; history of hypertension, diabetes, myocardial infarction, and stroke; smoking and alcohol consumption; family history of early-onset coronary artery disease; creatinine; triglycerides; high-density lipoprotein cholesterol; low-density lipoprotein cholesterol; cardiac troponin I; left ventricular ejection fraction; number of diseased vessels; emergency PCI for this episode; elective PCI for this episode; and discharge medication (aspirin, carbamazepine, statin, beta-receptor antagonist, angiotensin-converting enzyme inhibitor/angiotensin receptor blocker). Abbreviations: TMAO, trimethylamine oxide; MACE, major adverse cardiovascular event.

**Figure 2 nutrients-17-01664-f002:**
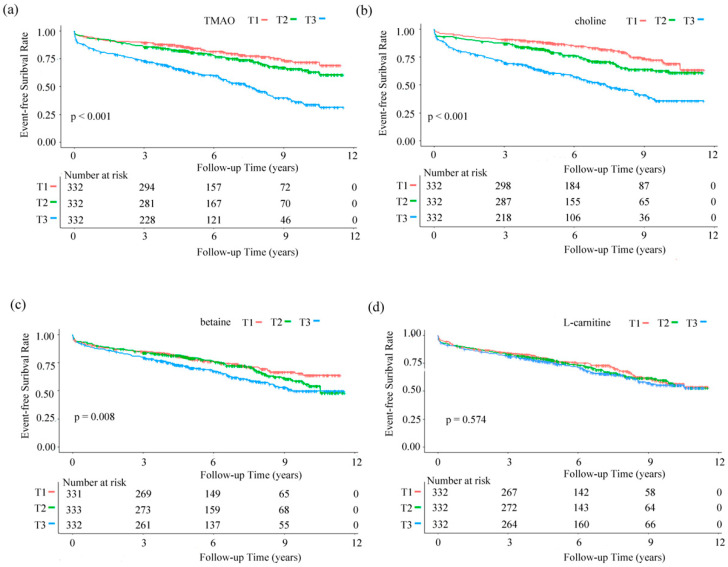
Kaplan–Meier curves for the risk of major adverse cardiovascular events. (**a**) TMAO, (**b**) choline, (**c**) betaine, and (**d**) L-carnitine. Horizontal coordinate, study follow-up time (years). Vertical coordinate, event-free survival rate. Abbreviations: TMAO, trimethylamine oxide; T1, tertile 1; T2, tertile 2; T3, tertile 3.

**Figure 3 nutrients-17-01664-f003:**
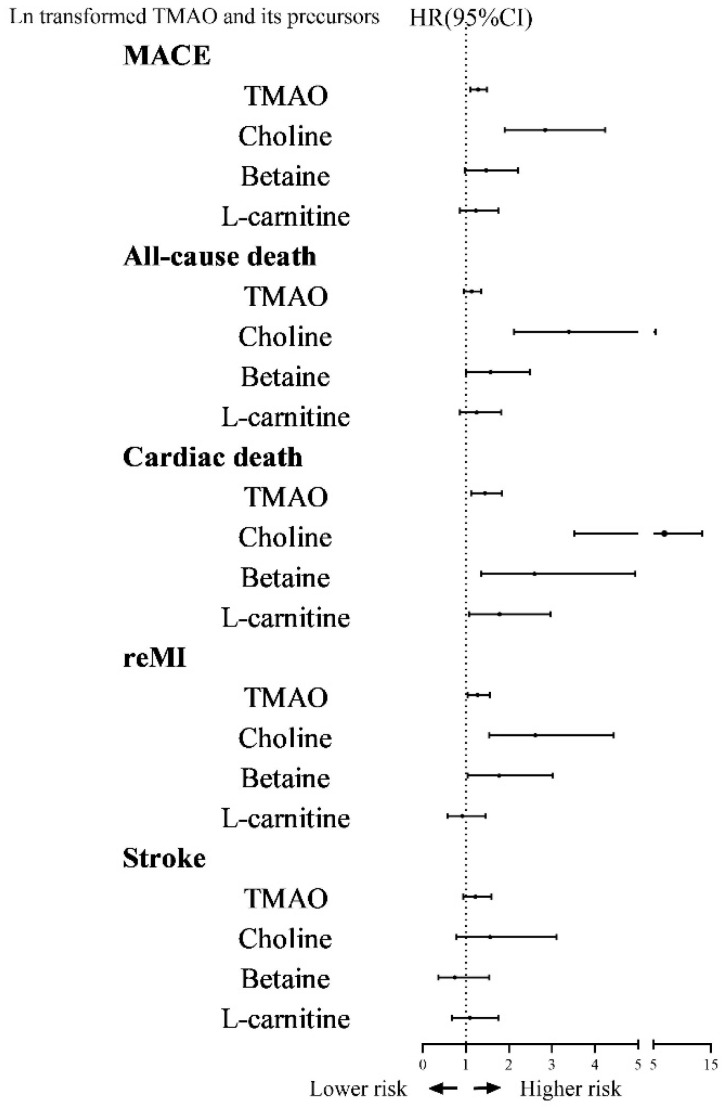
The relationship between TMAO, choline, betaine, and L-carnitine and each endpoint event in brief. Abbreviations: TMAO, trimethylamine oxide; MACE, major adverse cardiovascular event; reMI, recurrent myocardial infarction.

**Figure 4 nutrients-17-01664-f004:**
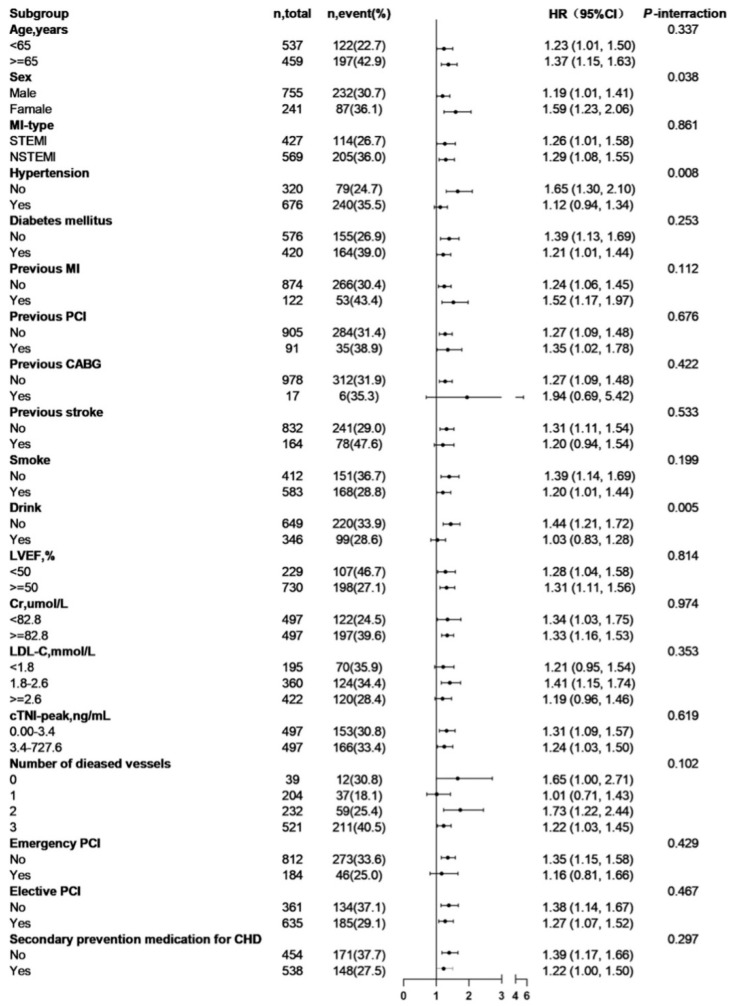
Interactions of risk factors with ln-transformed TMAO levels and MACE. Abbreviations: CABG, coronary artery bypass grafting; CI, confidence interval; Cr, creatinine; cTnI, cardiac troponin I; HR, hazard ratio; LDL-C, low-density lipoprotein cholesterol; LVEF, left ventricular ejection fraction; MACE, major adverse cardiovascular event; MI, myocardial infarction; NSTEMI, non-ST-elevation myocardial infarction; PCI, percutaneous coronary intervention; STEMI, ST-elevation myocardial infarction; CHD, coronary heart disease.

**Figure 5 nutrients-17-01664-f005:**
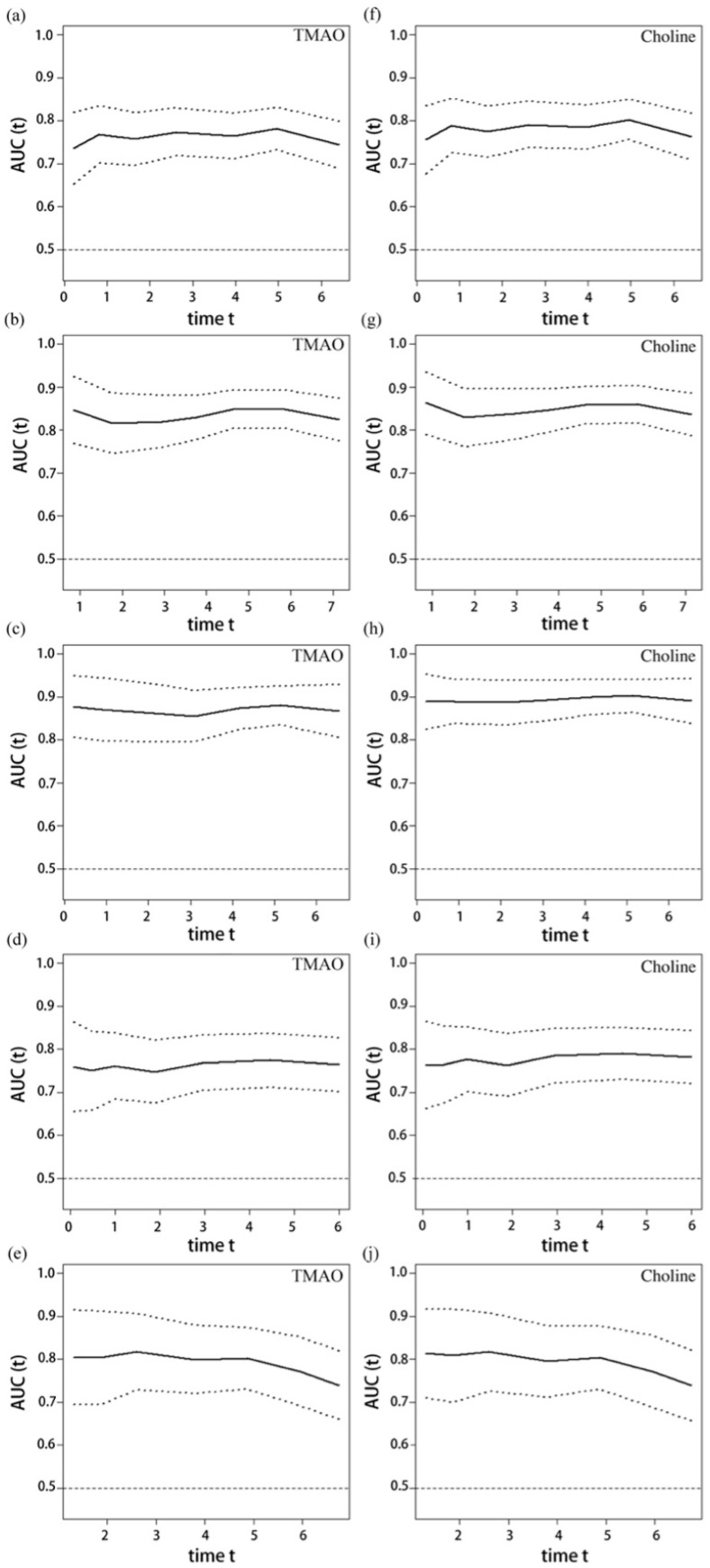
The time-dependent areas under the curve for TMAO, choline, and the various endpoint events. The solid line represents the fitting line, while the dashed line indicates the 95% confidence interval. (**a**–**e**) The time-dependent areas under the curve for ln-transformed TMAO versus MACE, all-cause mortality, cardiac death, recurrent infarction, and stroke, respectively. (**f**–**j**) The time-dependent areas under the curve for ln-transformed choline versus MACE, all-cause mortality, cardiac death, recurrent infarction, and stroke, respectively. Abbreviations: TMAO, trimethylamine-N-oxide; AUC, area under the curve.

**Figure 6 nutrients-17-01664-f006:**
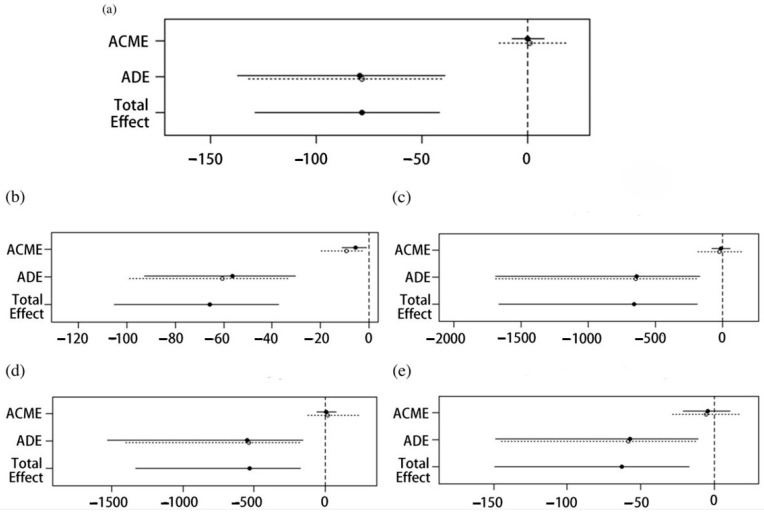
Analysis of intermediation effects. (**a**–**e**) Analysis of the mediating effect of choline on MACE, all-cause death, cardiac death, recurrent myocardial infarction, and stroke events when TMAO was used as a mediating effector, respectively. Abbreviations: ACME, average causal mediation effect; ADE, average direct effect.

**Table 1 nutrients-17-01664-t001:** Baseline characteristics of the study population.

Variables	Total (n = 996)	Non-MACE (n = 677)	MACE (n = 319)	*p* **-Value**
Male	755 (75.8)	523 (77.3)	232 (72.7)	0.120
Age (years)	63.3 ± 12.5	61.2 ± 12.2	67.8 ± 11.9	<0.001
MI type, n (%)				0.002
STEMI	427 (42.9)	313 (46.2)	114 (35.7)	
NSTEMI	569 (57.1)	364 (53.8)	205 (64.3)	
Medical history, n (%)				
Hypertension	676 (67.9)	436 (64.4)	240 (75.2)	<0.001
Diabetes mellitus	420 (42.2)	256 (37.8)	164 (51.4)	<0.001
Hyperlipidemia	582 (58.4)	402 (59.4)	180 (56.4)	0.378
Previous stroke	164 (16.5)	86 (12.7)	78 (24.5)	<0.001
Chronic kidney disease	110 (11.0)	44 (6.5)	66 (20.7)	<0.001
Old myocardial infarction	122 (12.3)	69 (10.2)	53 (16.6)	0.004
Previous PCI	91 (9.1)	56 (8.3)	35 (11.0)	0.168
Previous CABG	17 (1.7)	11 (1.6)	6 (1.9)	0.766
Personal history, n (%)				
Smoke	583 (58.6)	415 (61.4)	168 (52.7)	0.009
Drink	346 (34.8)	247 (36.5)	99 (31.0)	0.089
Laboratory indexes				
TMAO (μg/mL)	0.2 (0.1–0.3)	0.2 (0.1–0.3)	0.3 (0.2–0.5)	<0.001
Choline (μg/mL)	1.9 (1.6–2.5)	1.9 (1.5–2.3)	2.2 (1.7–2.8)	<0.001
Betaine (μg/mL)	5.3 ± 1.7	5.1 ± 1.6	5.6 ± 2.0	<0.001
L-carnitine (μg/mL)	6.1 ± 2.6	6.0 ± 2.1	6.4 ± 3.3	0.028
Creatinine (μmol/L)	82.8 (72.1–99.2)	80.7 (70.6–93.5)	91.1 (75.1–118.5)	<0.001
TCHO (mmol/L)	4.2 ± 1.1	4.3 ± 1.1	4.0 ± 1.1	<0.001
TG (mmol/L)	1.3 (1.0–2.0)	1.4 (1.0–2.1)	1.3 (1.0–1.8)	0.007
LDL-C (mmol/L)	2.6 ± 0.9	2.6 ± 0.9	2.4 ± 0.8	0.001
HDL-C (mmol/L)	1.0 ± 0.2	1.0 ± 0.2	0.9 ± 0.2	0.092
cTnI-peak (ng/mL)	3.4 (0.6–19.7)	3.3 (0.6–18.3)	4.2 (0.8–25.7)	0.246
BNP-peak (pg/mL)	230.0 (87.0–547.2)	171.0 (60.0–392.0)	455.0 (180.2–1031.0)	<0.001
hsCRP (mg/L)	4.1 (1.4–13.3)	3.9 (1.2–11.6)	5.6 (1.8–22.3)	<0.001
LVEF (%)	58.5 ± 13.0	60.4 ± 12.2	54.4 ± 13.6	<0.001
Angiography finding, n (%)				
Number of diseased vessels				<0.001
0	39 (3.9)	27 (4.0)	12 (3.8)	
1	204 (20.5)	167 (24. 7)	37 (11.6)	
2	232 (23.3)	173 (25.6)	59 (18.5)	
3	521 (52.3)	310 (45.8)	211 (66.1)	
Emergency PCI	184 (18.5)	138 (20.4)	46 (14.4)	0.024
Selected PCI	635 (63.8)	450 (66.5)	185 (58.0)	0.009
Medication, n (%)				
Aspirin	975 (98.3)	662 (98.1)	313 (98.7)	0.452
Ticagrelor	62 (6.3)	59 (8.7)	3 (1.0)	<0.001
Clopidogrel	863 (87.0)	574 (85.0)	289 (91.2)	0.007
ACEIs/ARBs	694 (70.0)	492 (72.9)	202 (63.7)	0.003
Beta-blocker	814 (82.1)	554 (82.1)	260 (82.0)	0.983
Statins	933 (94.2)	640(95.0)	293 (92.4)	0.114
Standard secondary prevention therapy	538 (54.2)	390 (57.8)	148 (46.7)	0.001

Data are shown as mean ± standard deviation (SD) or median (25–75th percentiles) for continuous variables and number (percentage) for categorical variables. Abbreviations: MI, myocardial infarction; STEMI, ST-segment elevation myocardial infarction; NSTEMI, non-ST-segment elevation myocardial infarction; PCI, percutaneous coronary intervention; CABG, coronary artery bypass grafting; TMAO, trimethylamine-N-oxide; TCHO, total cholesterol; TG, triglyceride; HDL-C, high-density lipoprotein cholesterol; LDL-C, low-density lipoprotein cholesterol; cTnI, cardiac troponin I; BNP, B-type natriuretic peptide; hsCRP, high-sensitivity C-reactive protein; LVEF, left ventricular ejection fraction; ACEIs/ARBs, angiotensin-converting enzyme inhibitors/angiotensin receptor blockers.

**Table 2 nutrients-17-01664-t002:** Association between TMAO levels and all endpoints.

Endpoint	Group	Event	Crude HR	Model 1	Model 2
		(n, %)	(95% CI)	HR (95% CI)	HR (95% CI)
MACE					
	Ln transform per 1 increase	319 (32.0)	1.65 (1.49, 1.82) **	1.28 (1.10, 1.49) *	1.19 (1.02, 1.39) *
	Tertile 1	68 (20.5)	1 (Ref)	1 (Ref)	1 (Ref)
	Tertile 2	90 (27.1)	1.31 (0.95, 1.79)	0.95 (0.68, 1.32)	0.93 (0.66, 1.29)
	Tertile 3	161 (48.5)	2.94 (2.21, 3.91) **	1.56 (1.13, 2.15) *	1.36 (0.98, 1.89)
	Trend test		<0.001	0.003	0.045
All-cause death					
	Ln transform per 1 increase	250 (25.1)	1.82 (1.62, 2.04) **	1.13 (0.95, 1.35)	1.03 (0.86, 1.23)
	Tertile 1	51 (15.4)	1 (Ref)	1 (Ref)	1 (Ref)
	Tertile 2	66 (19.9)	1.25 (0.87, 1.80)	0.70 (0.48, 1.03)	0.67 (0.45, 0.99) *
	Tertile 3	133 (40.1)	3.06 (2.22, 4.23) **	1.09 (0.75, 1.57)	0.89 (0.61, 1.31)
	Trend test		<0.001	0.389	0.817
Cardiac death					
	Ln transform per 1 increase	127 (12.8)	1.90 (1.63, 2.22) **	1.44 (1.12, 1.84) *	1.28 (0.99, 1.66)
	Tertile 1	21 (6.3)	1 (Ref)	1 (Ref)	1 (Ref)
	Tertile 2	34 (10.2)	1.57 (0.91, 2.71)	0.98 (0.55, 1.74)	0.92 (0.52, 1.65)
	Tertile 3	72 (21.7)	3.96 (2.44, 6.44) **	1.56 (0.90, 2.70)	1.23 (0.70, 2.16)
	Trend test		<0.001	0.066	0.366
reMI					
	Ln transform per 1 increase	185 (18.6)	1.64 (1.44, 1.87) **	1.27 (1.04, 1.55) *	1.21 (0.98, 1.49)
	Tertile 1	40 (12.0)	1 (Ref)	1 (Ref)	1 (Ref)
	Tertile 2	51 (15.4)	1.25 (0.83, 1.90)	0.93 (0.60, 1.45)	0.88 (0.57, 1.38)
	Tertile 3	94 (28.3)	2.79 (1.92, 4.03) **	1.48 (0.97, 2.27)	1.31 (0.84, 2.02)
	Trend test		<0.001	0.042	0.170
Stroke					
	Ln transform per 1 increase	107 (10.7)	1.54 (1.28, 1.86) **	1.22 (0.94, 1.59)	1.19 (0.90, 1.56)
	Tertile 1	21 (6.3)	1 (Ref)	1 (Ref)	1 (Ref)
	Tertile 2	31 (9.3)	1.45 (0.83, 2.53)	1.02 (0.57, 1.82)	1.03 (0.57, 1.84)
	Tertile 3	55 (16.6)	3.23 (1.95, 5.34) **	1.72 (0.98, 3.01)	1.68 (0.94, 2.99)
	Trend test		<0.001	0.035	0.057

Model 1: Sex; age; MI type; history of hypertension, diabetes, myocardial infarction, and stroke; smoking and alcohol consumption; family history of early-onset coronary artery disease; creatinine; triglyceride; high-density lipoprotein cholesterol; low-density lipoprotein cholesterol; cardiac troponin I; left ventricular ejection fraction; number of vessels; emergency PCI for this episode; elective PCI for this episode; and discharge medication (aspirin, ticagrelor, statins, beta-blocker, angiotensin-converting enzyme inhibitors/angiotensin receptor blockers). Model 2: model 1 + choline, betaine, and L-carnitine. Abbreviations: TMAO, trimethylamine-N-oxide; MACE, major adverse cardiovascular event; reMI, recurrent myocardial infarction; HR, hazard ratio; CI, confidence interval; *, *p* < 0.05; **, *p* < 0.001; other abbreviations as in Table 1.

**Table 3 nutrients-17-01664-t003:** Associations between choline levels and all endpoints.

Endpoint	Group	Event	Crude HR	Model 1	Model 2
		(n, %)	(95% CI)	HR (95% CI)	HR (95% CI)
MACE					
	Ln transform per 1 increase	319 (32.0)	5.39 (3.96, 7.33) **	2.84 (1.90, 4.23) **	2.82 (1.81, 4.38) **
	Tertile 1	68 (20.5)	1 (Ref)	1 (Ref)	1 (Ref)
	Tertile 2	90 (27.1)	1.48 (1.08, 2.01) *	1.35 (0.97, 1.88)	1.29 (0.92, 1.81)
	Tertile 3	161 (48.5)	3.14 (2.37, 4.16) **	1.91 (1.38, 2.65) **	1.76 (1.25, 2.48) *
	Trend test		<0.001	<0.001	0.001
All-cause death					
	Ln transform per 1 increase	250 (25.1)	8.02 (5.72, 11.24) **	3.39 (2.12, 5.40) **	3.36 (2.01, 5.60) **
	Tertile 1	51 (15.4)	1 (Ref)	1 (Ref)	1 (Ref)
	Tertile 2	66 (19.9)	1.39 (0.95, 2.03)	1.21 (0.81, 1.82)	1.18 (0.79, 1.78)
	Tertile 3	133 (40.1)	4.12 (2.98, 5.71) **	2.06 (1.39, 3.05) **	1.97 (1.32, 2.95) *
	Trend test		<0.001	<0.001	<0.001
Cardiac death					
	Ln transform per 1 increase	127 (12.8)	10.81 (6.89, 16.97) **	6.86 (3.51, 13.41) **	5.46 (2.61, 11.38) **
	Tertile 1	21 (6.3)	1 (Ref)	1 (Ref)	1 (Ref)
	Tertile 2	34 (10.2)	2.61 (1.39, 4.89) *	2.92 (1.45, 5.86) *	2.74 (1.36, 5.55) *
	Tertile 3	72 (21.7)	8.21 (4.64, 14.52) **	5.19 (2.64, 10.19) **	4.61 (2.31, 9.21) **
	Trend test		<0.001	<0.001	<0.001
reMI					
	Ln transform per 1 increase	185 (18.6)	4.59 (3.07, 6.86) **	2.61 (1.54, 4.43) **	2.53 (1.42, 4.52) *
	Tertile 1	40 (12.0)	1 (Ref)	1 (Ref)	1 (Ref)
	Tertile 2	51 (15.4)	1.45 (0.97, 2.17)	1.31 (0.84, 2.04)	1.25 (0.80, 1.95)
	Tertile 3	94 (28.3)	2.96 (2.05, 4.29) **	1.90 (1.23, 2.94) *	1.71 (1.09, 2.69) *
	Trend test		<0.001	0.003	0.017
Stroke					
	Ln transform per 1 increase	107 (10.7)	3.35 (1.92, 5.84) **	1.56 (0.78, 3.10)	1.92 (0.88, 4.17)
	Tertile 1	21 (6.3)	1 (Ref)	1 (Ref)	1 (Ref)
	Tertile 2	31 (9.3)	1.35 (0.82, 2.22)	1.26 (0.75, 2.11)	1.33 (0.78, 2.26)
	Tertile 3	55 (16.6)	2.22 (1.38, 3.56) **	1.25 (0.73, 2.15)	1.39 (0.78, 2.48)
	Trend test		<0.001	0.430	0.275

Model 1: Sex; age; MI type; history of hypertension, diabetes, myocardial infarction, and stroke; smoking and alcohol consumption; family history of early-onset coronary artery disease; creatinine; triglyceride; high-density lipoprotein cholesterol; low-density lipoprotein cholesterol; cardiac troponin I; left ventricular ejection fraction; number of vessels; emergency PCI for this episode; elective PCI for this episode; and discharge medication (aspirin, ticagrelor, statins, beta-blocker, angiotensin-converting enzyme inhibitors/angiotensin receptor blockers). Model 2: model 1 + TMAO, betaine, and L-carnitine. Abbreviations: TMAO, trimethylamine-N-oxide; MACE, major adverse cardiovascular event; reMI, recurrent myocardial infarction; HR, hazard ratio; CI, confidence interval; *, *p* < 0.05; **, *p* < 0.001; other abbreviations as in Table 1.

**Table 4 nutrients-17-01664-t004:** Analysis of the predictive effects of the choline model compared to the TMAO model.

		C Index (95% CI)	Continuous NRI	*p*-Value	IDI	*p*-Value
MACE	TMAO *	0.732 (0.702, 0.761)	0.097	0.100	0.021	0.066
	Choline *	0.744 (0.715, 0.772)				
All-cause death	TMAO *	0.814 (0.786, 0.843)	0.185	0.007	0.030	0.020
	Choline *	0.823 (0.796, 0.851)				
Cardiac death	TMAO *	0.853 (0.820, 0.886)	0.199	0.126	0.039	0.146
	Choline *	0.872 (0.843, 0.898)				
Recurrent myocardial infarction	TMAO *	0.742 (0.703, 0.781)	0.114	0.106	0.014	0.206
	Choline *	0.752 (0.714, 0.790)				
Stroke	TMAO *	0.750 (0.702, 0.798)	−0.031	0.997	−0.000	0.831
	Choline *	0.750 (0.702, 0.799)				

Other variables in the model: Sex; age; MI type; history of hypertension, diabetes, myocardial infarction, PCI and CABG, and stroke; smoking and alcohol consumption; creatinine; TG; HDL-C; LDL-C; cardiac troponin I (cTnI); left ventricular ejection fraction; number of vessels; emergency PCI and elective PCI for this episode; and discharge medication (aspirin, tegretol, statin, beta-receptor antagonist, angiotensin-converting enzyme inhibitors/angiotensin receptor blockers). *, ln-transformed. Abbreviations: TMAO, trimethylamine oxide; MACE, major adverse cardiovascular event; NRI, net reclassification improvement; IDI, integrated discrimination improvement.

## Data Availability

The original contributions presented in this study are included in the article and Appendix A. Further inquiries can be directed to the corresponding authors.

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
