# Peer review of "Relationship Between Level of Trimethylamine Oxide and the Risk of Recurrent Cardiovascular Events in Patients with Acute Myocardial Infarction"

_nutrients, 2025, doi:10.3390/nu17101664_

Round 1
Reviewer 1 Report
Comments and Suggestions for Authors
The study of Wenjun et al. provides interesting insights into the links between metabolites, particularly TMAO and choline, and cardiovascular events. The results are very interesting. However, certain points need to be clarified and other points should be considered and highlighted as limitations/weaknesses of the study.
- Data were collected up until December 2021. The authors do not address the effect of the COVID-19 pandemic on mortality of patients included in this study. Do the collected data allow for the exclusion of mortality associated with COVID-19 complications, or complication that increase CVD-mortality or cardiovascular events? If not, how should this be considered or discussed?
- The low proportion of patients who received standard secondary prevention treatment in the MACE group could influence the occurrence of cardiovascular events. The authors do not appear to have taken this into account in their analysis.
- Although the study adjusted for some traditional risk factors, comorbidities and prior treatments of the patients could be significant confounding factors. For example, could the difference in treatment between the groups (particularly the use of ticagrelor, statins, and secondary prevention treatments) explain, at least in part, the difference in the occurrence of cardiovascular events between the groups?
- It is worth noting that the follow-up period (6.4 years) is relatively short. A longer follow-up period could provide more robust information on the recurrence of cardiovascular events and the long-term effects of the measured biological factors.
Author Response
Comments 1: Data were collected up until December 2021. The authors do not address the effect of the COVID-19 pandemic on mortality of patients included in this study. Do the collected data allow for the exclusion of mortality associated with COVID-19 complications, or complication that increase CVD-mortality or cardiovascular events? If not, how should this be considered or discussed?
Response 1: Thanks for raising the question regarding the impact of the COVID-19 pandemic on our study. The COVID-19 pandemic overlapped with our research period from 2020 to 2021 and may influence the all-cause mortality. However, its impact could be considered negligible, as China experienced extremely low life expectancy losses, COVID-19 mortality, and excess mortality. [Lancet, 2024, 403(10440): 2100-2132/Lancet Reg Health West Pac, 2024, 43: 100795.]. Based on the above explanations, we believe that our cohort has been affected by COVID-19, but its impact on mortality can be considered minimized. Therefore, it is not necessary to prematurely terminate the analysis of follow-up data at an earlier stage that predates the COVID-19 pandemic. We appreciate your insightful comments and the opportunity to clarify the association between COVID-19 and our research.
Comments 2: The low proportion of patients who received standard secondary prevention treatment in the MACE group could influence the occurrence of cardiovascular events. The authors do not appear to have taken this into account in their analysis.
Response 2:
The proportion of patients receiving standard secondary prevention treatment in the MACE group of this study population was indeed lower than that in the non-MACE group. However, this was adjusted for in the outcome analysis, as indicated in the annotation of model 1 in the table. Meanwhile, thanks for your reminder. When we rechecked the data, we found that the two sets of data in Table 1 were reversed during the input process. We have already corrected it in the latest version and marked it in the table.
Comments 3: Although the study adjusted for some traditional risk factors, comorbidities and prior treatments of the patients could be significant confounding factors. For example, could the difference in treatment between the groups (particularly the use of ticagrelor, statins, and secondary prevention treatments) explain, at least in part, the difference in the occurrence of cardiovascular events between the groups?
Response 3:
As our study is retrospective, there are specific differences between the two groups in terms of traditional risk factors, comorbidities, and the patients' previous treatments. However, these differences have been adjusted for in the analysis. Therefore, the influence of comorbidities and the patients' previous treatments on the results has been considered.
Comments 4: It is worth noting that the follow-up period (6.4 years) is relatively short. A longer follow-up period could provide more robust information on the recurrence of cardiovascular events and the long-term effects of the measured biological factors.
Response 4:
As you mentioned, a longer follow-up period could provide more robust information on the recurrence of cardiovascular events and the long-term effects of the measured biological factors. However, in previous studies, recurrent cardiovascular events following acute myocardial infarction mostly occurred within the first year[Eur Heart J. 2023 Oct 12;44(38):3720-3826.]. This is also a factor that we took into consideration when setting the follow-up duration.
Reviewer 2 Report
Comments and Suggestions for Authors
The introduction is fine. You should summarize key points more concisely regarding the background and emphasize the knowledge gap this study fills.
Materials and methods are described in detail.
How was bias minimized in endpoint verification?
In results, ensure all variables and statistical terms are clearly explained in captions. You could also add a visual or table to quickly compare TMAO, choline, betaine, and L-carnitine across outcomes.
In discussion you could contrast choline vs. TMAO findings more explicitly and briefly compare with non-Chinese cohorts (e.g., EPIC, CARDIA studies).
Author Response
Reviewer2:
Comments 1: The introduction is fine. You should summarize key points more concisely regarding the background and emphasize the knowledge gap this study fills.
Materials and methods are described in detail.
Response 1:
Thank you for your suggestion. We have revised the final part of the background section in the latest version of the manuscript. We have summarized the key points and emphasized the knowledge gaps this study fills.
Comments 2: How was bias minimized in endpoint verification?
Response 2:
To ensure the accuracy and objectivity of the endpoint events, especially death, and minimize bias, we adopted a multiple validation approach, including telephone follow-up and matching with external databases. This approach was also mentioned in the section on cardiovascular outcomes.
Comments 3: In results, ensure all variables and statistical terms are clearly explained in captions. You could also add a visual or table to quickly compare TMAO, choline, betaine, and L-carnitine across outcomes.
Response 3:
Thanks for your constructive opinions. We have rechecked the results to ensure all variables and statistical terms are clearly explained in captions. The deficiencies were revised in the latest manuscript. A quick comparison of TMAO, choline, betaine, and L-carnitine across outcomes was added in Figure 2.
Comments 4: In discussion you could contrast choline vs. TMAO findings more explicitly and briefly compare with non-Chinese cohorts (e.g., EPIC, CARDIA studies).
Response 4:
The EPIC and CARDIA studies are two significant TAMO-related studies based on non-Chinese community populations. However, our study mainly focuses on the population suffering from AMI. During our discussion, a comparison was made with the CARDIA study (lines 168-182), and the EPIC study was also mentioned in our background (reference 8). If further discussion is needed, please let us know.
Reviewer 3 Report
Comments and Suggestions for Authors
The study included 996 AMI patients with a median follow-up of 6.4 years, which strengthens reliability and longitudinal inference. Simultaneous measurement of TMAO, choline, betaine and L-carnitine by LC-MS/MS allows integrated assessment of metabolites.
Points for clarification:
While it has been claimed that choline pathways are TMAO-independent, no experimental validation has been presented.
Dietary habits (e.g., consumption of seafood or red meat), gut microbiota composition, and adherence to post-discharge medications were not considered.
Although adjusted for creatinine, GFR was not used, which may have underestimated the influence of the kidneys on TMAO values.
Author Response
The study included 996 AMI patients with a median follow-up of 6.4 years, which strengthens reliability and longitudinal inference. Simultaneous measurement of TMAO, choline, betaine and L-carnitine by LC-MS/MS allows integrated assessment of metabolites.
Points for clarification:
Comments 1: While it has been claimed that choline pathways are TMAO-independent, no experimental validation has been presented.
Response 1:
Thanks for your valuable suggestion,in our study, it was found that choline could better predict recurrent cardiovascular events in patients with myocardial infarction than TMAO. Moreover, through mediation effect analysis, it was discovered that the predictive ability of choline might be independent of TMAO. Similar findings have also been reported in previous studies conducted among community populations. Therefore, we made such speculation in the discussion, but it is only a possibility. More rigorously designed studies are needed to verify this. We also revised the ambiguous expressions in the discussion. (line 157)
Comments 2: Dietary habits (e.g., consumption of seafood or red meat), gut microbiota composition, and adherence to post-discharge medications were not considered.
Response 2:
The factors you mentioned are all very important confounding factors that might affect the outcome of our research. However, it is a pity that we did not collect the relevant information. Your opinion is also included as a limitation of our research in the latest version of our manuscript. (lines 249-251)
Comments 3: Although adjusted for creatinine, GFR was not used, which may have underestimated the influence of the kidneys on TMAO values.
Response 3:
In previous studies, eGFR was used to adjust renal function, while in our study, the variable for adjustment was serum creatinine. Under your suggestion, we also conducted a statistical analysis to adjust eGFR. The results of the re-statistical analysis showed no significant difference in choline outcome compared with that adjusting creatinine. Meanwhile, although baseline TMAO level in patients with AMI were associated with the risk of MACE, but not cardiac death, and recurrent MI. The statistics related to TMAO and choline are presented in the following table (R1 and R2). Thank you again for your suggestions.
Reviewer 4 Report
Comments and Suggestions for Authors
This is an interesting and important study on trimethylamine oxide (TMAO) and myocardial infarction in humans.
From the paper, it is unclear the relationships between TMAO and other molecules studied including choline, betaine, and L-carnitine. If authors describe how TMAO is synthesized from choline, betaine, and L-carnitine more in detail with some chemical structural information, it would be helpful to the readers.
It would be interesting to the readers if the authors elaborate more about choline, betaine, and carnitine being abundant in various types of seafood, dairy products, egg yolk, and meat. Does it mean that people should not eat seafood, dairy products, egg yolk, and meat?
Author Response
This is an interesting and important study on trimethylamine oxide (TMAO) and myocardial infarction in humans.
Comments 1: From the paper, it is unclear the relationships between TMAO and other molecules studied including choline, betaine, and L-carnitine. If authors describe how TMAO is synthesized from choline, betaine, and L-carnitine more in detail with some chemical structural information, it would be helpful to the readers.
Response 1:
Thank you for your suggestion. The relationships between TMAO and other molecules is elaborated in detail in References 2 and 3. Background also contains some specific descriptions that are conducive to readers' understanding (The second paragraph in the background section).
Comments 2: It would be interesting to the readers if the authors elaborate more about choline, betaine, and carnitine being abundant in various types of seafood, dairy products, egg yolk, and meat. Does it mean that people should not eat seafood, dairy products, egg yolk, and meat?
Response 2:
Thank you for your suggestion. However, the results of this study are only applicable to a specific group of people after myocardial infarction. The study itself is retrospective and cannot provide suggestions for dietary structure adjustment. Further designing a large-scale randomized controlled study might guide the adjustment of nutritional habits of myocardial infarction patients while they are undergoing the existing secondary prevention.
Round 2
Reviewer 4 Report
Comments and Suggestions for Authors
.